# Tumor-to-Tumor Metastasis: Dissemination of Cutaneous Squamous Cell Carcinoma Involving Parotid Warthin Tumor—Case Report

**DOI:** 10.3390/diseases12070140

**Published:** 2024-07-01

**Authors:** Michał Gontarz, Krystyna Gałązka, Krzysztof Gąsiorowski, Jakub Bargiel, Tomasz Marecik, Paweł Szczurowski, Grażyna Wyszyńska-Pawelec

**Affiliations:** 1Department of Cranio-Maxillofacial Surgery, Jagiellonian University Medical College, 30-688 Cracow, Poland; krzysztof.gasiorowski@uj.edu.pl (K.G.); jakub.bargiel@uj.edu.pl (J.B.); tomasz.marecik@uj.edu.pl (T.M.); pawel.szczurowski@uj.edu.pl (P.S.); grazyna.wyszynska-pawelec@uj.edu.pl (G.W.-P.); 2Department of Pathology, Jagiellonian University Medical College, 30-688 Cracow, Poland; krystyna.galazka@uj.edu.pl

**Keywords:** tumor-to-tumor metastasis, Warthin tumor, secondary tumors, cutaneous squamous cell carcinoma

## Abstract

Warthin tumors (WTs) are the second most common salivary gland tumors, most commonly found in the tail of the parotid gland. The lymphoid components of a WT may also serve as a site for tumor-to-tumor metastasis. This report describes the metastasis of cutaneous squamous cell carcinoma (cSCC) from the preauricular region to a parotid WT. A 68-year-old male patient was admitted to the Department of Cranio-Maxillofacial Surgery of the Jagiellonian University in Cracow, Poland, with a two-year history of a growing, painless skin tumor of the right preauricular region. The patient was eligible for surgical treatment with elective neck dissection at the Va, III, II, I levels with parotidectomy and the excision of the cSCC. In the histopathological examination of the components of the surgical specimen beside the primary cutaneous squamous cell carcinoma (cSCC), a parotid WT was found; in the majority, it was occupied and destroyed by the metastatic cSCC and radically removed. After a tumor board consultation, due to the advanced stage (pT3N2b LVI2 PNI0 R0), with three intraparotid lymph node metastases and LVI2, the patient was authorized for postoperative radiotherapy. In conclusion, tumor-to-tumor metastasis is an extremely rare entity. Surgery is the standard of care for both WTs and head and neck cSCC. In most cases, metastasis into the parotid gland can be successfully treated with superficial parotidectomy with facial nerve preservation. Dissemination into the parotid gland requires elective neck dissection, as well as adjuvant treatment.

## 1. Introduction

Warthin tumors (WTs) are the second most common salivary gland tumors, most commonly found in the tail of the parotid gland. A WT is almost exclusively limited to the parotid gland [1,2,3,4]. Most cases involve the inferior pole, although 10% occur in the deep lobe. Occasionally, tumors arise in the adjacent lymph nodes. The extraparotid location of these tumors is uncommon and is observed in up to 8% of all WTs. Abnormal FDG uptake in PET/CT is typical for WTs and may suggest metastatic lymph nodes, especially when occurring synchronously with head and neck cancer [5,6]. 

Microscopically, WTs are composed of two components varying in proportions from case to case: a true neoplastic oncocytic bilayered epithelium growing in cystic, solid, papillary patterns and reactive stromal lymphoid tissue, often with lymph follicles present. The epithelial and lymphoid components of WTs may; however, very rarely, they undergo malignant transformation [7]. The lymphoid component of a WT may also serve as a site for tumor-to-tumor metastasis. Tumor-to-tumor metastasis is an extremely rare phenomenon, especially in the head and neck region [8]. Only a few cases have been published in which parotid WTs have been the recipient of distant metastases [9,10].

This report describes the metastasis of cutaneous squamous cell carcinoma (cSCC) from the preauricular region to a parotid WT.

## 2. Case Report

In December 2023, a 68-year-old male patient was admitted to the Department of Cranio-Maxillofacial Surgery of the Jagiellonian University in Cracow, Poland, with a two-year history of a growing, painless skin tumor of the right preauricular region. Progression in the last six months was not observed. In the outpatient clinic, an incisional biopsy of the tumor was performed, which revealed the infiltration of squamous cell carcinoma. 

The patient had experienced many comorbidities, including chronic renal failure, ischemic heart disease, a history of NSTEMI myocardial infarction in July 2020, tachycardia with a broad QRS complex, chronic heart failure with a reduced ejection fraction, chronic obstructive pulmonary disease (COPD), hypothyroidism, ongoing treatment for bladder cancer with BCG therapy, and arterial hypertension. In addition, the patient had undergone pacemaker implantation, stent graft placement for an abdominal aortic aneurysm, and a double urethrocystoscopy with the transurethral resection of a bladder tumor (TURBT).

Clinically, a 3 cm exophytic skin cancer was observed in the right preauricular region. Moreover, a non-painful tumor approximately 3 cm in diameter was identified in the lower pole of the parotid gland, suspected for metastasis (Figure 1). Facial nerve function remained intact.

Due to ongoing renal insufficiency, a non-contrast CT scan was performed and showed a 26 × 21 mm exophytic tumor in the preauricular region and a 30 × 20 mm oval soft tissue lesion in the parotid tail anterior to the sternoclavicular muscle, suggestive of a metastatic node (Figure 2). There were no other enlarged nodes in the neck besides those noted. Distant metastases were excluded on chest and abdominal CT. After consultation with the tumor board, the patient was authorized for selective neck dissection at levels I–III with superficial parotidectomy and the excision of the cSCC.

## 3. Surgery

The patient was eligible for surgical treatment with elective neck dissection at the Va, III, II, I levels with parotidectomy and the excision of the cSCC. During the surgery, intraoperative neuromonitoring was used to identify the facial nerve branches (Inomed C2 Nerve Monitor). The superficial lobe of the parotid gland with the metastatic tumor was removed. Following a superficial parotidectomy, the clinical inspection of the deep parotid lobe revealed no enlarged or suspected foci. This implied a contraindication to the extension of the surgery to a total parotidectomy. The lymph nodes adjacent to the external jugular vein were also resected. The defect of the preauricular region was reconstructed by a rotated bilobed cervical flap. The active drainage was removed on the fourth postoperative day. The patient was discharged home six days after the surgery. The healing process was uneventful, apart from the mild paresis of the marginal branch of the right facial nerve.

## 4. Pathology

The histopathological examination of the surgical skin specimen revealed extensive infiltrative cutaneous squamous cell carcinoma G2 (intermediate histologic grade of differentiation), with minimal focal keratinization only. The maximum depth of invasion (DOI) was approximately 14.2 mm, and no nerve infiltration was found (perineural invasion PNI0), whereas the invasion of the small lymph vessels on the periphery and in the vicinity of the carcinoma infiltrate was present (lymphovascular invasion LVI2). The surgical cancer excision was radical, with adequate margins > 5 mm (R0). 

In the histopathological examination of the remaining components of the surgical specimen, a parotid Warthin tumor was found; in the majority, it was occupied and destroyed by the metastatic squamous carcinoma (with the largest metastasis dimension of about 24 mm) (Figure 3, Figure 4 and Figure 5) and radically removed. A second small parotid Warthin tumor, not involved in the cancer metastasis, was removed nearby. Additionally, a cancer metastasis 10.7 mm in diameter in the intraparotid lymph node and a small lymph node with cancer metastasis 1.4 mm in diameter were found, without extranodal extension (ENE-), and were also completely removed. The examination of 12 lymph nodes in the neck specimen did not reveal metastases; thus, the final pathological stage established, according to the 2017 AJCC classification, was pT3N2b (ENE-) with LVI2, PNI0, and R0.

## 5. Follow-Up

After the tumor board consultation, due to the advanced stage (pT3N2b LVI2 PNI0 R0), with three intraparotid lymph node metastases and LVI2, the patient was authorized for postoperative radiotherapy. No locoregional recurrence was observed during the 6-month follow-up. However, the slight paresis of the marginal branch of the right facial nerve remained (Figure 6).

## 6. Discussion

The WT, or cystadenolymphoma of the parotid gland, was first described by the Germans Albrecht and Arzt in 1910 and named by the American pathologist Aldred Scott Warthin in 1929 [11]. Importantly, the incidence of Warthin tumors has increased in recent years. This may be due to an increased life expectancy, increased body mass index (BMI), smoking habits, and improved diagnostic tools and imaging sensitivity [1,12]. In oncological patients, especially on PET/CT scans, a WT may be found as an incidental tumor [5,6]. In addition, a WT is characterized by the possibility of multifocal occurrence within one parotid gland or bilaterally. A WT is classified as a benign, epithelial salivary gland tumor. However, malignant forms of WTs can be found in the literature. Seifert et al., in 1977, defined different forms of carcinoma in WTs, such as secondary malignancies in pre-existing benign WTs, carcinoma metastasis within benign WTs, the coincidence of WTs with other tumors, and questionable primary malignant WTs, which have not been described yet [7].

### 6.1. Secondary Malignancies in Pre-Existing Benign WT

The malignant transformation of a benign WT is rare, occurring in less than 1% of cases. Various types of carcinoma arising from the epithelial component of a WT have been reported in the English literature. Squamous cell carcinoma and mucoepidermoid carcinoma are the most common. Other types of carcinoma arising in WTs are oncocytic carcinoma, undifferentiated carcinoma, and adenocarcinoma, not otherwise specified (NOS) [7,13]. The diagnosis of the malignant transformation of a benign WT to carcinoma is based on several criteria proposed by Seifert: (1) the exact evidence of a pre-existing benign WT; (2) a continuous transition from a benign double-layered oncocytic epithelium to a clearly malignant epithelium; (3) the infiltrative growth of the carcinoma into the surrounding lymphoid stroma and adjacent tissue; (4) the development of regional metastasis in the cervical lymph nodes; (5) the exclusion of metastasis to the lymphoid stroma from an extrasalivary primary carcinoma [7]. Moreover, the lymphoid component of a WT may, albeit rarely, undergo malignant transformation into malignant lymphoma associated with a WT. The most common are non-Hodgkin lymphomas (ca. 80%), of which follicular lymphoma and diffuse large B cell lymphoma (DLBCL) are usually observed [14].

### 6.2. Carcinoma Metastasis within Benign WT

Tumor-to-tumor metastasis is a rare occurrence. According to the Campbell criteria, tumor-to-tumor metastasis is described as follows: (1) there must be at least two primary tumors with the recipient being a true neoplasm; (2) direct ingrowth or tumor emboli must be excluded; and (3) tumor metastasis to the lymph nodes affected by lymphoproliferative disease must be excluded [15]. While the last two criteria are met, there is debate as to whether a WT is a true neoplasm or not. According to the 2022 WHO classification, a WT is classified as a benign epithelial tumor. Current molecular studies suggest that a subset of Warthin tumors are characterized by a recurrent t(11;19) and associated CRTC1-MAML2 fusion oncogene, supporting a clonal origin in such cases. Furthermore, CRTC1-MAML2 is a frequent feature of mucoepidermoid carcinoma [16]. 

Petraki et al. reviewed the incidence of recipient tumors and found that renal cancer was the most common, followed by sarcomas, meningiomas, thyroid neoplasms, and pituitary adenomas. The most common primary donor tumor is lung cancer, followed by breast, prostate, and thyroid carcinomas [8]. 

According to the literature, regional metastases to the parotid WT are most commonly from cSCC and malignant melanoma [17,18]. In contrast, distant metastasis to WTs occurs sporadically. Only four cases of distant metastasis have been described. The donor tumors were due to lung, renal, and breast cancer [9,10,19]. However, our literature review did not show any regional metastasis to WTs. The case of a malignant WT described by De la Pava et al. in 1965 could be an example of the regional metastasis of skin cancer to a WT due to the fact that the patient had a history of radiotherapy for skin cancer of the ear and lower lip [20]. For this reason, our article is believed to be the first well-documented case report of a patient with regional cSCC metastasis to a WT of the parotid gland. Among the 152 patients (175 WTs) operated on in our department from 2000 to 2023, no malignant transformation of a WT was observed, and only one case (0.57%) of regional cSCC metastasis to a WT, described in this article, was found. 

### 6.3. Coincidence of WT with Other Tumors

The synchronous coincidence of WTs with other tumors within the same parotid gland can be observed in some cases. These tumors can be either benign or malignant. Malignant tumors may represent a primary epithelial salivary cancer or metastasis to the parotid gland [7]. In our cohort, one case of the co-occurrence of a WT with a pleomorphic adenoma was observed, as well as two cases of cSCC infiltrating the parotid gland with WTs. In all of these cases, the WTs were classified as incidentalomas. Even when cancer infiltration is present on the synchronous WT, such cases cannot be classified as carcinomas in WTs.

cSCC is the most frequently observed type of metastatic tumor in the parotid gland [21,22,23]. The nodal metastasis of cSCC is mainly related to the parotid region and II neck levels, which are also typical sites for WTs [24,25,26,27]. The National Comprehensive Cancer Network (NCCN) 2024 guidelines recommend superficial parotidectomy with ipsilateral neck dissection when indicated [28]. However, elective neck dissection (END) appears necessary in all cN0 neck patients with confirmed parotid metastases from cSCC. Furthermore, the extent of elective neck dissection (END) should be determined by the site of the primary lesion. In cases of facial primaries, such as this one, the END should include at least levels II and III [24]. Additionally, in cases of cutaneous malignancies with metastases to the parotid gland, the superficial lymph nodes of the external jugular vein must be removed. The method of treatment remains unchanged when cSCC metastasizes to the parotid gland’s WT. Metastasis to a WT in the parotid gland is treated in the same way as lymph node metastasis and is an indication for adjuvant radiotherapy.

## 7. Conclusions

In conclusion, tumor-to-tumor metastasis is an extremely rare entity in the head and neck region. To our knowledge, this is the first fully documented case of the regional metastasis of cSCC involving a parotid WT. Surgery is the standard of care for both WTs and cSCC metastasis to the parotid gland. cSCC metastasis to the parotid gland can be successfully treated by superficial parotidectomy with facial nerve preservation. However, dissemination to the parotid gland requires additional elective neck dissection, as well as adjuvant treatment.

## Figures and Tables

**Figure 1 diseases-12-00140-f001:**
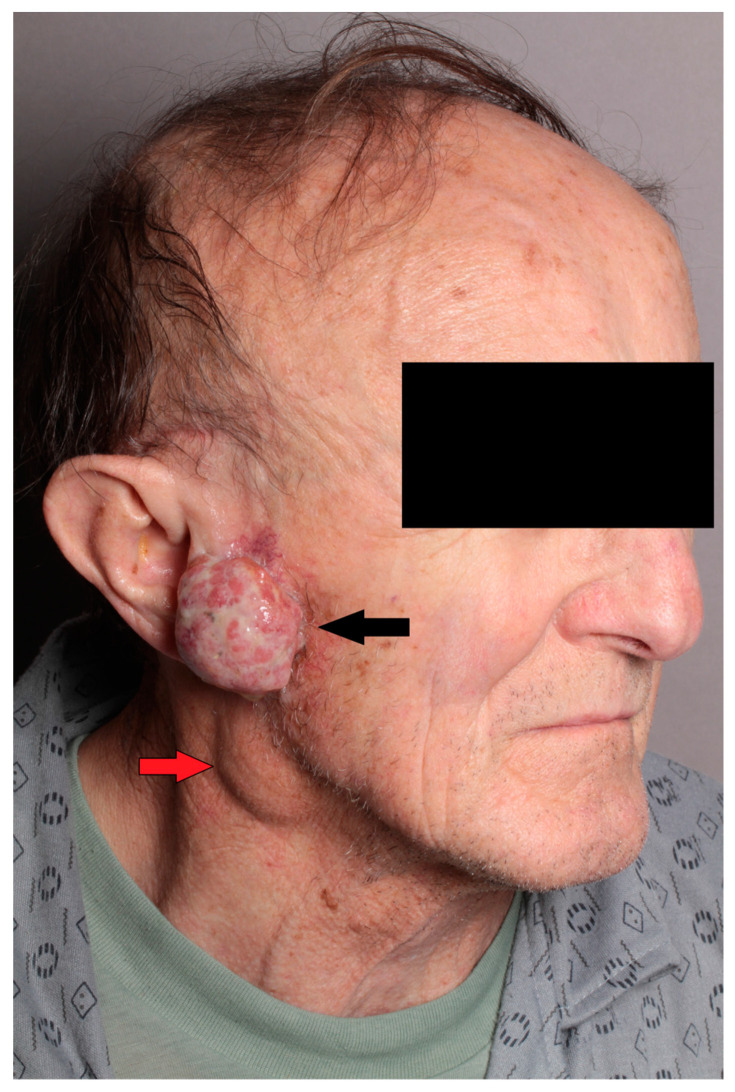
Clinical image of the patient showing exophytic SCC of the skin of the preauricular region on the right side (black arrow), with metastasis in the tail of the parotid gland (red arrow).

**Figure 2 diseases-12-00140-f002:**
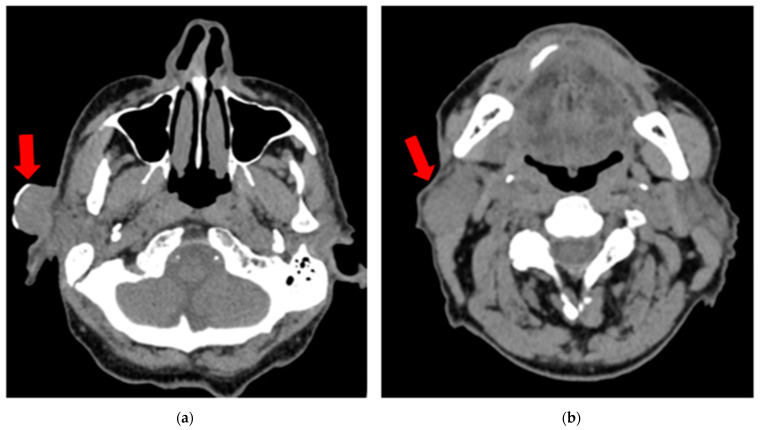
Non-contrast CT scan (vertical view) showing (**a**) a 26 × 21 mm exophytic tumor (cSCC) in the preauricular region (red arrow); (**b**) a 30 × 20 mm oval soft tissue lesion in the parotid tail anterior to the sternoclavicular muscle, suggestive of a metastatic node (red arrow).

**Figure 3 diseases-12-00140-f003:**
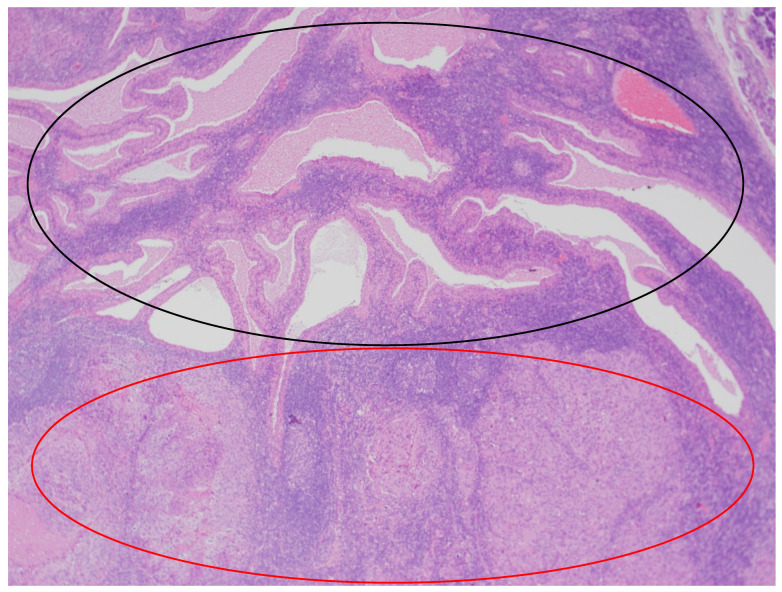
The parotid Warthin tumor, with the benign epithelial cystic component in the superior part of the photo (black circle) and squamous cell carcinoma metastasis in the inferior part (red circle). HE staining. Magn. ×88.

**Figure 4 diseases-12-00140-f004:**
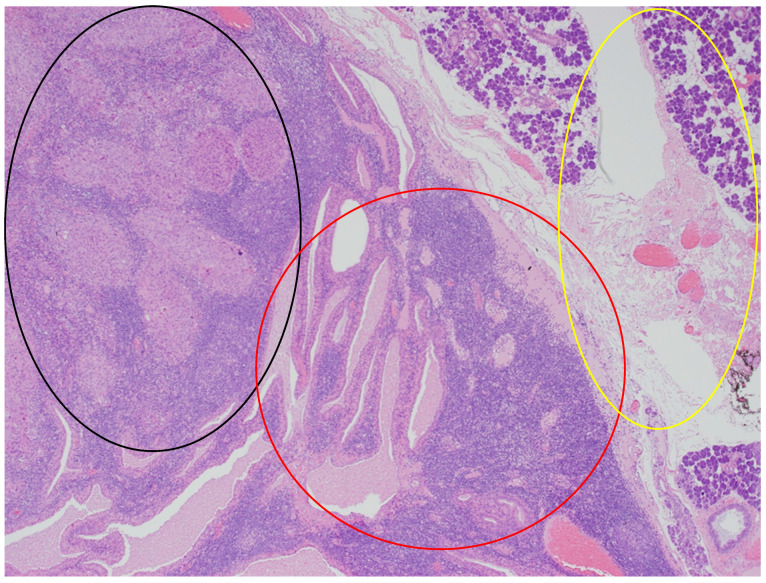
The parotid Warthin tumor, with the benign epithelial cystic component in the central and right parts of the photo (red circle), with the rim of parotid gland tissue not involved in any neoplasm on the right (yellow circle), and the squamous cell carcinoma metastasis in the superior left part (black circle). HE staining. Magn. ×88.

**Figure 5 diseases-12-00140-f005:**
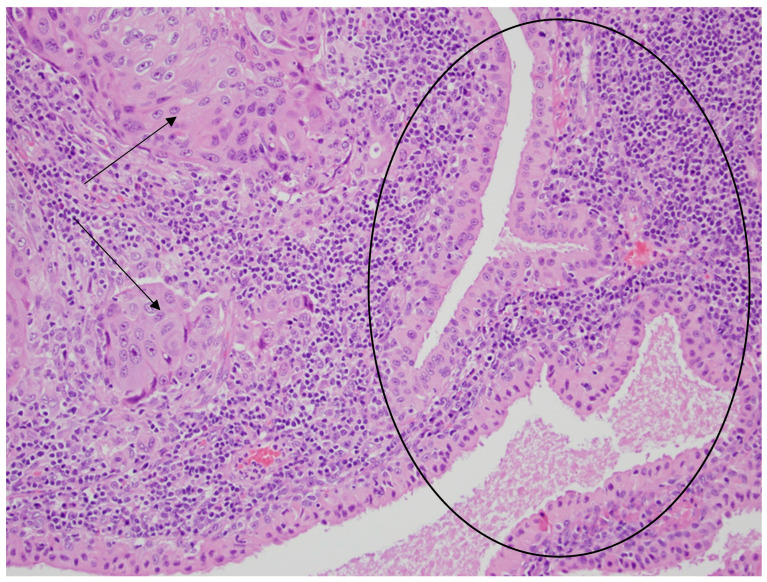
Glandular cystic structures lined by bland oncocytic cells of the Warthin tumor in the central and right parts of the photo (black circle) and the nests of metastatic squamous carcinoma cells in the lymphoid stroma (the left part of the photo—black arrows). HE staining. Magn. ×440.

**Figure 6 diseases-12-00140-f006:**
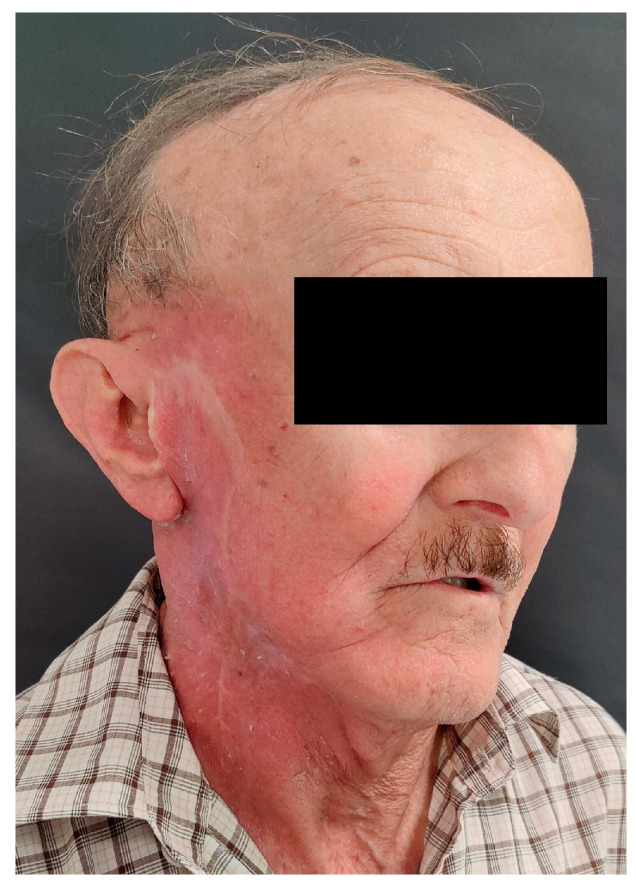
Clinical image of the patient 6 months after surgery and PORT, with visible radiation reactions on facial skin.

## Data Availability

Restrictions apply to the availability of these data. The data were obtained from patients treated at the Department of Cranio-Maxillofacial Surgery, Cracow, Poland, and cannot be shared in accordance with the General Data Protection Regulation (EU) 2016/679.

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
