# Peer review of "Tumor-to-Tumor Metastasis: Dissemination of Cutaneous Squamous Cell Carcinoma Involving Parotid Warthin Tumor—Case Report"

_diseases, 2024, doi:10.3390/diseases12070140_

Round 1
Reviewer 1 Report
Comments and Suggestions for Authors
The present manuscript reports a case of metastasis of cutaneous squamous cell carcinoma from the preauricular region to the a parotid Warthin tumor. The case is well described and appropriately illustrated.
There is one major problem which reaches beyond semantics as it deals with essentials of metastasis. The authors mention several types of metastasis, namely lymph node metastasis (L 128), regional metastasis (L161 & 183) and distant metastasis (L 184). The term regional metastasis is not common and needs a clear definition and criteria, including a description of the route of spread (Lymphatic? Hematogenous?). They also mention “regional metastasis in lymph nodes” (L 161). What is the difference between lymph node metastasis and regional metastasis? Does the literature provide examples of regional metastasis outside lymph nodes? Misquotations of the literature add to the confusion. The authors provide the first case of regional metastasis to the WT (L 190). This is in contrast with L183 and the references (17 & 18) quoted there: Alevi et al. (2014), and Gunduz et al. (1999) report on a carcinoma arising in a WT and neither of both considers this lesion as metastasis.
Details:
One. The four cases of distant metastases are from ref [9 ](2 cases), ref [10] (1 case) and ref [8] (2 cases) but not ref [19], which is not about metastasis (L 186). Replace [19] by [8]?
Two. Label pictures showing all structures mentioned in the legend (L70; L83; L 117; L 120; L124), on behalf of the non-expert reader. Explain abbreviations that are not mentioned in ajcc staging manual (L 115).
Author Response
We would like to express our gratitude to you for taking the time to review and comment on our article. In response to the main problem posed by the term "regional metastasis," we believe that the content in the article is correct and does not require clarification. For a surgeon or an oncologist treating head and neck cancer, "regional metastasis" is defined as metastasis that localizes to the head and neck (that is, above the clavicle). In contrast, distant metastases represent metastases below the clavicle. Furthermore, whether the metastasis originates from the lung to the salivary gland or a metastasis from the salivary gland to the lung, these are always distant metastases that arise through the hematogenous route. In contrast, metastasis from the salivary gland to the cervical lymph nodes represents regional metastasis via the lymphatic route. Head and neck cancers (90% SCC) metastasize primarily via the lymphatogenous route to the regional lymph nodes of the neck. Distant metastasis (via hematogenous route) accounts for less than 10% of cases. The terms cervical lymph node metastasis or regional metastasis can therefore be used interchangeably, as they refer to the same phenomenon. In-transit metastasis of head and neck melanoma can also be observed in the field of close proximity metastasis.
Another problem mentioned by the Reviewer concerns to the article No 20 (line 190)
De la Pava, S., Knutson, G. H., Mukhtar, F., & Pickren, J. W. (1965). Squamous cell carcinoma arising in Warthin's tumor of the parotid gland: First case report. Cancer, 18, 790–794. https://doi.org/10.1002/1097-0142(196506)18:6<790::aid-cncr2820180617>3.0.co;2-1
The article describes the first case of SCC arising in WT. However, due to patients clinical history we supposed that this cancer in WT could be a regional metastasis from lip cancer. This information was included in discussion as a suspicion of cSCC metastasis to WT.
Also in the article No 19: Smith and Fesmire, the authors, had similar concerns. They described:
“De la Pava and associates discuss a case of so-called squamous-cell carcinoma arising in a Warthin’s tumor in the parotid gland. In this case, the patient, had received radiation therapy to the skin of the ear for a basal-cell carcinoma and to the lower lip for a basosquamous carcinoma in 1951. From the discussion, this seems to be a case of squamous-cell carcinoma metastatic to involved lymphoid tissue in the gland which was superimposed upon a benign papillary cystadenoma lymphomatosum with some alteration due to the effects of the tumor.”
The next problem is information about distant metastasis to parotid WT. The description and references regarding distant metastasis to parotid WT (4 cases) in our article are accurate and correct. Explanation:
Reference No 8 Petraki et. al.
The article described 2 cases tumor-to-tumor metastasis in the abdomen, not metastasis to parotid WT.
Case 1
“Focus of the carcinoma of the uterine cervix in the renal cell carcinoma-tumor-to-tumor metastasis.”
Case 2
“Foci of the urothelial carcinoma of the urinary bladder in the solitary fibrous tumor of the pleura-tumor-to-tumor metastasis.”
On the other hand, the article No 19 Smith and Fesmire described the case of lung metastasis (distant metastasis) to parotid WT.
“Bronchogenic carcinoma in patient with papillary cystadenoma lymphomatosum.”
In conclusion, the literature describes two cases of lung cancer (No. 9 and No. 19), one case of breast cancer (No. 10), and one case of renal cancer metastasis to parotid WT.
Regarding the photos the suggested changes have been made.
Reviewer 2 Report
Comments and Suggestions for Authors
The report describes the metastasis of cutaneous squamous cell 15 carcinoma (cSCC) from the preauricular region to the parotid WT. A 68-year-old male patient was 16 admitted to the Department of Cranio-Maxillofacial Surgery of the Jagiellonian University in Cra-17 cow, Poland, with a two-year history of a growing, painless skin tumor of the right preauricular 18 region.
The case report was well written, also describing an interesting content. Minor points:
I would recommend signaling the locations described in the legends also in the images (for example, with arrows) for figures 3,4 and 5. Maybe figure 3, 4 and 5 could be concentrated in one figure with three panels (Panel A, B and C) and legend written in accordance.
The back strap in the face of the patient could be a bit bigger to avoid identification.
Author Response
Thank you for your comments. Regarding the photo showing the clinical image of the patient, the suggested changes have been made. Also, photographs 3-5 showing the histological image of the surgical specimen were changed with additional marks.
However, compressing three photographs into a single collage will make the image less readable. In truth, the histological image is the most important evidence of cSCC metastasis in the WT.
Round 2
Reviewer 1 Report
Comments and Suggestions for Authors
I have no further comments.